# Morphological, Physiological, and Biochemical Impacts of Different Levels of Long-Term Water Deficit Stress on *Linum album* Ky. ex Boiss. Accessions

**Reza Kiani [1], Vahideh Nazeri [1,\*], Majid Shokrpour [1] and Christophe Hano [2]** 

[1] Department of Horticultural Sciences, College of Agriculture & Natural Resources, University of Tehran, P.O. Box 4111, Karaj 3158777871, Iran; kianireza37@ut.ac.ir (R.K.); shokrpour@ut.ac.ir (M.S.)

[2] Laboratoire de Biologie des Ligneux et des Grandes Cultures, INRAE USC1328, University of Orleans, CEDEX 2, 45067 Orléans, France; hano@univ-orleans.fr

\* Correspondence: nazeri@ut.ac.ir

**Abstract:** *Linum album* (Ky. ex Boiss.) is an important medicinal plant that produces compounds such as the well-known anticancer lignan podophyllotoxin and fatty acids. Despite its high medicinal value, it has not yet been studied in detail under agricultural conditions. This study was conducted to evaluate the morphological, phenological, and physiological responses of six *L. album* accessions under different levels of water deficit treatments (100%, 75%, 50%, and 25% available water) in pot conditions. Based on the results, some of the morphological characteristics of the response to water deficit were established. Accessions UTLA7, UTLA9, and UTLA10 showed a higher seed yield and dry weight of the vegetative part. There was a substantial difference in the occurrence of phenological stages in the accessions. The maturation process was accelerated in plants under stress conditions, and accession UTLA9 completed its complete growth cycle faster than the other accessions. The physiological responses of the different accessions did not show the same pattern on the basis of the characteristics studied, and significant differences were observed depending on the trait and accession. Among the most important results of this study was the diversity of responses in different accessions. Based on these results, it is recommended that morphological features (such as seed yield per plant, plant height, number of inflorescences per plant, shoot and root dry weight) be used to select tolerant accessions for the desired product.

**Keywords:** abiotic stress; *Linum album* Ky. ex Boiss.; morphological properties; phenology; pigments; diversity

## 1. Introduction

*Linum album* Ky. ex Boiss. is a perennial medicinal plant belonging to the Linaceae family. This species is an endemic plant in Iran, where it grows in the northwest, west, and central regions. The flowering and maturing stage of this plant lasts from May to July [1,2]. *L. album* contains important lignan compounds such as podophyllotoxin and 6-methoxypodophyllotoxin, which have antiviral and antitumoral properties [3]. Podophyllotoxin, a well-known lignan, serves as the unique starting compound for the semisynthesis of the leading anticancer drug Etoposide (VP16) and its derivatives (class of topoisomerase II inhibitors) used in a dozen anticancer chemotherapy treatments [4,5]. These drugs are on the list of essential medicines of the World Health Organization (WHO) [5]. However, the availability of podophyllotoxin is restricted because it is still exclusively extracted from the rhizomes of *Podophyllum* plants growing in wild forests in Asia. The supply of

*Podophyllum* plants is rather limited since the occurrence of these plant species is scarce and they require a long growth period (five to seven years) before harvest [5,6]. Nowadays, this species is endangered by overcollection, which exceeds its regeneration capacity, and a lack of cultivation. Consequently, *Podophyllum* is listed in Appendix II of the Convention on International Trading of Endangered Species (CITES), and Etoposide has been identified by the French Drug Agency (ANSM, Agence National de Securité du Médicament) on the list of products "out of stock" several times since late 2016 [5]. Chemical synthesis of podophyllotoxin is difficult due to the presence of four contiguous chiral centers and the presence of a base-sensitive *trans*-lactone moiety. As neither chemical production nor extraction from in vitro plant cultures is economically competitive with the extraction of podophyllotoxin from *Podophyllum* roots, alternatives are sought such as other natural sources [5]. In recent decades, several putative alternative sources of podophyllotoxin have been identified, including *Cupressaceae*, *Lamiaceae*, *Linaceae*, *Podophyllaceae*, and *Polygalaceae* [5]. Some *Linum* species are considered to be a promising alternative source of podophyllotoxin given their lignan accumulation capacities [5–10]. The *Linum* genus comprises more than 230 species, largely distributed among temperate and subtropical climates [5], and species from the *Sylinum* section, including *L. album*, have been reported to accumulate high amounts of podophyllotoxin and its derivatives in their aerial parts, roots, and seeds [5,11,12]. *L. album* seeds constitute some of the richest alternative sources of podophyllotoxin and its derivatives [11,12]. *L. album* seeds also accumulate fatty acid compounds such as palmitic, stearic, oleic, linoleic, and linolenic acid [13]. However, contrary to its congener, the common flax *L. usitatissimum*, little is known about the agronomical performance of *L. album*, in particular its response to drought stress.

Plants face a large number of biotic (fungi, viruses, and insects) and abiotic (drought and salinity) stress in their environment. Environmental stresses confine crop yield and create many changes in their molecular processes such as variation in metabolite profile [14]. Growth is complete through cell division, enlargement, and differentiation and depends on genetic, physiological, ecological, and morphological events and their complex interactions. The quality and quantity of plant growth depend on these events, which are affected by water deficit. Water deficit stress is a condition of insufficient water availability, caused by intermittent to continuous periods without irrigation [15]. In Iran, more than 75% of the arid and semiarid regions have been classified as water-deficient regions [16]. The outcome of water deficit is limitations in the distribution and survival of plants in arid and semiarid regions [17–19]. These areas include most areas where *L. album* grows [1]. Water deficit condition, the most important stress in plants, conducts an unusual increase in reactive oxygen species (ROS) production [20]. ROS can damage the cell membrane and increase the production of malondialdehyde (MDA) content [21]. Another important that effect inhibits the growth and photosynthetic abilities of plants is the loss of balance between the production of ROS and the antioxidant defense [22,23], which causes the accumulation of ROS, leading to oxidative damage to proteins, membrane lipids, and other cellular components [24]. Plants can use both enzymatic and nonenzymatic systems to control ROS production. The enzymatic antioxidant processes involved the activity of enzymes such as catalase (CAT), superoxide dismutase (SOD), ascorbate peroxidase (APX), and polyphenol oxidase (PPO) [20,25,26]. The antioxidant defense system consists of nonenzymatic components, such as ascorbate, glutathione, proline, glycine betaine, amino acids, and/or phenolic compounds [20]. Water deficit stress also affects the rate of plant growth and development [27,28]. Under water stress conditions, plants complete their life cycle faster than under normal conditions; consequently, crop growth stages have a short duration, with fewer days to accumulate assimilates during the life cycle, and the production of biomass is reduced [29–31]. Crops have a definite temperature requirement before they attain certain phenological stages. Accumulative heat units and systems were adopted for determining the dates of the flowering and maturity of different field crops [28,32]. However, susceptibility to drought varies as a function of the phenological stage, depending on plant species and genotypes, and considerable inter- and intraspecific variations can be observed [33,34]. Abscisic acid (ABA) is a key phytohormone involved in the control of many

physiological processes such as seed dormancy and germination and in the response to many abiotic and biotic signals [35,36]. ABA is known as a key stress-signaling hormone, acting in the regulation of stomatal closure, the synthesis of compatible osmolytes, and in the upregulation of genes, leading to adaptive responses [35]. Proline is an active protective osmolyte for plants formed as a result of oxidative stress by free-radical stimulation [37,38] in response to different environmental stresses [39,40]. Glycine betaine is another essential compatible solute found in plants, animals, and bacteria in response to water deficit stress [29,41–43]. Many studies have shown that glycine betaine plays an important role in improving plant tolerance under many abiotic stresses, including drought stress [44]. In addition to the direct protective functions of glycine betaine, either by beneficial effects on enzyme and membrane integrity or as a compatible solute, it can also indirectly protect cells from environmental stress by participating in signal transduction pathways [45].

To date, most of the studies focusing on *L. album* have been designed to enhance lignan compounds in in vitro cultures. Various techniques, such as optimizing the culture medium [46], the use of elicitors [47], or inducing polyploidy [48], have been successful in increasing the accumulation of lignans in various *L. album* in vitro culture. However, insufficient research has been conducted under greenhouse or field conditions. The main objective of the present study is to investigate and understand the morphological, phenological, and physiological responses of six different *L. album* accessions to different levels of water deficit stress (100%, 75%, 50%, and 25% available water) in pot conditions.

## 2. Materials and Methods

### 2.1. Plant and Soil Materials

This study was a continuation of a project on morphological and physiological variations in different populations of *L. album* in the west of Iran [13,49]. To study the effect of water deficit stress on some valuable characteristics of this species, six superior accessions were selected and subjected to the treatments (Table 1, seeds of accession UTLA12 were obtained from the seed gene bank at the Forest and Rangeland Research Institute in Tehran, Iran).

**Table 1.** *L. album* seed source locations.

| Accession Code | Latitude (N) | Longitude (E) | Altitude (m) | Voucher Number [a] |
|:---:|:---:|:---:|:---:|:---:|
| UTLA1 | 34°13′56″ | 48°57′25″ | 1904 | 6426 |
| UTLA6 | 34°55′50″ | 48°11′34″ | 2176 | 6425 |
| UTLA7 | 34°41′12″ | 48°38′02″ | 2124 | 6430 |
| UTLA9 | 34°46′11″ | 48°43′17″ | 1955 | 6427 |
| UTLA10 | 34°22′45″ | 48°40′02″ | 1721 | 6428 |
| UTLA12 | 32°54′11″ | 50°4′39″ | 2630 | - |

[a] Department of Horticultural Sciences Herbarium, University of Tehran.

Seeds were treated with 1000 ppm gibberellic acid for 24 h to overcome seed dormancy and germinated in a plastic germination tray containing coco-peat in April 2018 [49]. The growing media consisted of a mixture of field soil, sand, and leaf mold (formed from decaying leaves to improve soil structure and water retention) in an equal ratio (Table 2).

**Table 2.** Edaphic parameters of soil.

| pH | EC (Ds/m) | Silt (%) | Clay (%) | Sand (%) | Soil CLASS | FC (%) | PWP (%) | OC (%) | Total N (%) | Usable K (mg/kg) | Usable P (mg/kg) |
|:---:|:---:|:---:|:---:|:---:|:---:|:---:|:---:|:---:|:---:|:---:|:---:|
| 8.1 | 2.8 | 21 | 11 | 68 | Sandy loam | 24.98 | 12.52 | 3.18 | 0.25 | 397 | 47.6 |

The leaf mold was provided by the botanic garden of the University of Tehran. Sixty days after germination, uniformly sized seedlings were randomly selected and transplanted into the pots (one seedling per pot). After a one-year growth period, in January 2019, the dried shoots of plants

were uniformly cut at 1 cm above soil level. All steps were performed in open-field conditions at the College of Agriculture and Natural Resources, University of Tehran (daily temperature conditions are provided in Table S1).

## 2.2. Water Stress Treatment

Irrigation of plants during the first year of cultivation was performed continuously to the extent of field capacity. Water deficit stress (different levels of irrigation consisting of 100 (control), 75, 50, and 25% of plant available water (AW = FC − PWP)) was applied in the second year of cultivation. The weight method was used, and the pots were weighed every 48 h. Due to nonuniformity in the growth of plants, flower bud emergence was considered the criterion for the onset of water deficit stress. The experiment was performed as a factorial experiment in a randomized complete block design (RCBD) with three replications and three observations (three pots) in each replication.

## 2.3. Determination of Phenological Stages

Phenological stages of accessions were recorded individually from the beginning of the growth of the first plant in the second year of growth (2 March 2019). The occurrence of phenological stages was reported based on growing degree days (GDD). The following formula [49] was used to calculate the GDD [50].

$$GDD = \sum [(Tmax - Tmin)/2] - Tbase \text{ with } Tbase = 2.67$$

## 2.4. Relative Water Content

Relative water content (RWC) was determined by the procedure outlined by Turner [51]. From each sample in the water stress treatments, 15 fully extended leaves were removed from the plant stem, weighed (FW), and floated on double distilled water for 24 h at 4 °C. Turgid leaves were quickly weighed, the weight of the samples was considered the turgidity weight (TW). The samples were oven-dried at 70 °C for 48 h and reweighed to obtain the dried leaf weight (DW). The relative water content was calculated by the following equation.

$$RWC \text{ (\%)} = [(FW - DW)/(TW - DW)] \times 100$$

## 2.5. Chlorophyll and Carotenoid Contents

For all physiological experiments, samples were taken from fully mature leaves during the maturing period. Samples from each replicate and treatment were separately mixed, ground, and stored at −80 °C.

To measure leaf chlorophyll content, 500 mg of the frozen powdered sample was mixed with 10 mL of 95% ethanol. The homogenized sample mixture was centrifuged at 8000 rpm for 15 min. Supernatants were read by a microplate spectrophotometer (BioTek Eon, Winooski, VT, USA) at 664 nm for chlorophyll-a (Ch a), 649 for chlorophyll-b (Ch b), and 470 nm for carotenoids [52]. The amounts of chlorophyll and carotenoid were calculated by the following formulas.

$$Ch \text{ a } (\mu g/mL) = 13.36_{A664} - 5.19_{A649}$$

$$Ch \text{ b } (\mu g/mL) = 27.43_{A649} - 8.12_{A664}$$

$$Carotenoids \text{ } (\mu g/mL) = (1000A470 - 2.13Ch \text{ a} - 97.63Ch \text{ b})/209$$

## 2.6. Determination of Proline and Glycine Betaine

Proline content was determined using the method established by Bates et al. [53]. A 500 mg amount of leaves was commixed in 10 mL of 3% sulfosalicylic acid, and the mixture was centrifuged at 10,000 rpm for 10 min. Then, 2 mL of the supernatant was added to 2 mL of an acid-ninhydrin solution

and 2 mL of glacial acetic acid in a tube. The tubes were incubated in a bain-marie at 100 °C for 1 h. The reaction was stopped in ice. The reaction mixture was extracted with 4 mL of toluene and vortexed for 15–20 s. The tubes were allowed to stand for at least 20 min in darkness at room temperature for the separation of toluene from the aqueous phase. The toluene phase was then collected in tubes, and the absorbance at 520 nm was measured with a microplate spectrophotometer (BioTek Eon, Winooski, VT, USA). The proline concentration was determined according to the standard curve of proline.

The amount of glycine betaine was evaluated according to Grieve and Grattan [54]. A 250 mg amount of leaf powder (dried leaves) was shaken with 10 mL of deionized water for 48 h at 25 °C. The extracts were filtered using filter paper, diluted (1:1) with $H_2SO_4$ (2N), and cooled in ice water for 60 min. A 0.2 mL volume of cold KI-I2 was then added to the samples and softly mixed. The tubes were kept at 4 °C for 16 h and centrifuged at 10,000 rpm for 15 min at 0 °C. The supernatant was attentively discarded, and periodide crystals were dissolved in 9 mL of 1,2-dichloroethane. After 2 h, the value of absorbance at 365 nm was evaluated using a spectrophotometer. The amount of glycine betaine was calculated according to the standard curve of glycine betaine.

## 2.7. Measurement of Electrolyte Leakage and Malondialdehyde Contents

Malondialdehyde determination began by homogenizing 500 mg of fresh leaves in 5 mL of 10% trichloroacetic acid (TCA). In the next step, samples were centrifuged at 12,000 rpm for 10 min at 4 °C. A 2 mL volume of supernatant was added to 4 mL of 0.6% thiobarbituric acid (TBA, in 10% TCA) and incubated at 100 °C in a bain-marie for 15 min. The samples were cooled at room temperature, and the absorbance of the supernatant was measured at 450, 532, and 600 nm using a microplate spectrophotometer (BioTek Eon, Winooski, VT, USA). The MDA contents were calculated by the following formula [55]:

$$\text{MDA } (\mu\text{mol g}^{-1} \text{ FW}) = 6.45 \, (A_{532} - A_{600}) - 0.56 \, A_{450}$$

Electrolyte leakage (EL) was estimated using a conductivity meter. Fresh leaf samples were cut into 10 pieces of equal sizes, each with a 4.5 mm diameter. The samples were then dipped in 15 mL of distilled water and shaken at 100 rpm for 24 h at room temperature. The initial electrical conductivity (EC1) of the solution was recorded. The tubes were then autoclaved at 120 °C for 20 min. After cooling the tubes to room temperature, the final EC (EC2) was recorded [56]. Finally, the percentage of ion leakage was calculated by the following equation:

$$\text{EL (\%)} = [EC1/EC2] \times 100$$

## 2.8. Enzymatic Antioxidant Activity

For enzyme assays, frozen leaf samples were ground to a fine powder with liquid nitrogen and extracted with 50 mM phosphate buffer (pH = 7.0). The extracts were centrifuged at 4 °C for 15 min at 13,000 rpm. The supernatant was then collected and used for the protein content assay and enzyme activities. Protein extraction was performed according to Bradford [57] using bovine serum albumin as a standard. Catalase activity was determined using the spectrophotometric method (BioTek Eon, Winooski, VT, USA) according to Hadwan [58]. Total guaiacol peroxidase activity was determined according to Plewa et al. [59]. The reaction mixture included 3000 μL of 50 mM phosphate buffer (pH = 7), 10 μL of 30% hydrogen peroxide, 3 μL of 200 μM guaiacol solution, and 100 μL of enzymatic extract. The addition of enzyme extract started the reaction, and the increase in absorbance was recorded at 470 nm for 4 min (Perkin Elmer, Waltham, MA, USA; UV-VIS Spectrophotometer, LAMBDA EZ201). The activity of ascorbate peroxidase was measured according to Ranieri et al. [60]. The reaction mixture contained 600 μL of 0.1 mM EDTA, 1500 μL of 50 mM phosphate buffer (pH = 7), 400 μL of 0.5 mM ascorbic acid, 400 μL of 30% hydrogen peroxide, and 100 μL enzyme extract. Enzyme activity assays

were recorded at 470 nm for 4 min (Perkin Elmer, Waltham, MA, USA; UV-VIS Spectrophotometer, LAMBDA EZ201, USA).

*2.9. Abscisic Acid (ABA) Extraction and Quantification*

A 250 mg amount of dried leaf samples (sampling was performed at the end of flowering stages) was extracted in 2 mL of aqEtOH (80%) solvent. The extracts were sonicated by an ultrasonic bath for 60 min (USC1200TH, Prolabo, Fontenay-sous-Bois, France) with a maximal heating power of 400 W (i.e., acoustic power of 1 W/cm$^2$). The extract supernatant was filtered through 0.45 μm nylon syringe membranes. ABA was quantified using a Phytodetek ABA ELISA kit (Agdia) and (±) *cis-trans* ABA (Sigma-Aldrich, Saint-Quentin Fallavier, France) as a standard.

*2.10. Statistical Analysis*

All data were subjected to analysis of variance (ANOVA) using SAS V.9.2 software followed by an LSD test with $p < 0.01$ as the significant differences between means. The results are presented as means ± SE (standard errors). Pearson's correlation was investigated with IBM SPSS Statistics 23.0. Cluster analysis was performed by Ward's method. Factor analysis was investigated by the principal components extraction method (factor rotation was performed by Varimax with the Kaiser normalization method) using the IBM SPSS Statistics 23.0 software (Chicago, IL, USA). The principal component analysis triplot was drawn based on the factor score of the first 3 components.

## 3. Results

*3.1. Morphological Properties*

The response of the six *L. album* accessions to different irrigation levels showed a significant difference in most traits, including the seed width, weight of 1000 seeds, inflorescence length, main branch length, plant height, leaf length, flower diameter, root length, and fresh and dry weight of shoots and roots (Table S2).

As shown in Table 3, under normal irrigation conditions (100% AW), the highest seed yield per plant was observed in accession UTLA1 (1.86 g) and the lowest was observed in accessions UTLA10 and UTLA12 (1.19 and 1.24, respectively). However, the weight of 1000 grains in accession UTLA1 was the lowest (3.12) and higher in accessions UTLA10 and UTLA12 (3.67 and 3.81, respectively) than other accessions. The number of flowers per inflorescence in accession UTLA9 (12.81) was significantly higher than other accessions. Plant height in accessions UTLA9 and UTLA12 (36.5 and 36.17) was higher than the other accessions. The highest shoot and root dry weight was observed in accessions UTLA9 and UTLA10, respectively, whereas accession UTLA1 showed the lowest shoot and root dry weight with a significant difference.

Under severe water stress treatment (25% AW), the highest seed yield per plant was seen in accession UTLA9 (1.9 g), and UTLA7 showed the lowest yield (0.5). The weight of 1000 seeds was the highest in accession UTLA10 (3.86) and the lowest in UTLA9 (2.98). Interestingly, the sample that showed the highest seed yield per plant under normal conditions and severe stress had the lowest 1000-seed weight, which indicates that the seeds were smaller in size but larger in number.

As observed in plant height, the number of flowers per inflorescence in the 25% AW treatment decreased in all accessions compared to the control. Accession UTLA9 showed the highest plant height (30.5 cm). The dry weight of the shoots and roots in accessions UTLA10 and UTLA12 was higher than in the other accessions. Accession UTLA1 had the lowest dry weight of shoots and roots under control conditions and the lowest weight under severe stress levels.

**Table 3.** Effect of water deficit stresses on the morphological traits of *L. album* accessions.

| Accession/ Irrigation | Number of Seeds per Capsule | Seed Yield per Plant (g) | Seed Width (mm) | Weight of 1000 Seeds (g) | Number of Flowers in Inflorescence | Number of Mature Capsules | Inflorescence Length (cm) | Length of the Main Branch (cm) | Plant Height (cm) |
|---|---|---|---|---|---|---|---|---|---|
| **UTLA1** | | | | | | | | | |
| 100% AW | 8.7 ± 0.15 | 1.86 ± 0.03 | 2.35 ± 0.03 | 3.12 ± 0.1 | 7.33 ± 0.51 | 6.76 ± 0.57 | 13.24 ± 0.38 | 14.83 ± 0.44 | 27.5 ± 0.76 |
| 75% AW | 8.31 ± 0.18 | 0.61 ± 0.04 | 2.29 ± 0.06 | 3.03 ± 0.14 | 11.67 ± 0.67 | 11 ± 1 | 13.33 ± 1.2 | 13.33 ± 0.67 | 23.83 ± 0.6 |
| 50% AW | 8.5 ± 0.69 | 0.6 ± 0.06 | 2.25 ± 0.03 | 3.12 ± 0.16 | 10.08 ± 1.08 | 9.42 ± 1.39 | 15.11 ± 0.49 | 11 ± 0.58 | 24.67 ± 2.19 |
| 25% AW | 7.56 ± 0.34 | 0.71 ± 0.06 | 2.31 ± 0.01 | 3.36 ± 0.16 | 8.08 ± 0.65 | 6.25 ± 0.8 | 17.75 ± 0.43 | 13.5 ± 1.04 | 25 ± 0.5 |
| **UTLA6** | | | | | | | | | |
| 100% AW | 7.55 ± 0.62 | 1.34 ± 0.16 | 2.43 ± 0.06 | 3.42 ± 0.16 | 11.39 ± 1.52 | 10.56 ± 1.44 | 15.37 ± 1.81 | 15.5 ± 1.32 | 29.33 ± 1.45 |
| 75% AW | 7.92 ± 0.52 | 0.96 ± 0.09 | 2.36 ± 0.04 | 3.51 ± 0.12 | 9.33 ± 0.95 | 8.83 ± 0.98 | 17.49 ± 1.5 | 17 ± 1.04 | 34.67 ± 0.73 |
| 50% AW | 7.04 ± 0.61 | 1.01 ± 0.1 | 2.48 ± 0.06 | 3.87 ± 0.13 | 12.18 ± 0.84 | 11.16 ± 0.63 | 15.48 ± 1.58 | 14.67 ± 0.73 | 30.17 ± 1.64 |
| 25% AW | 9.06 ± 0.48 | 1.33 ± 0.09 | 2.55 ± 0.01 | 3.7 ± 0.12 | 9.94 ± 1.06 | 8.44 ± 0.29 | 12.59 ± 0.6 | 15.83 ± 0.6 | 27.83 ± 1.3 |
| **UTLA7** | | | | | | | | | |
| 100% AW | 8.49 ± 0.18 | 1.47 ± 0.15 | 2.46 ± 0.02 | 3.57 ± 0.15 | 8.94 ± 0.78 | 8.49 ± 0.7 | 16.45 ± 0.78 | 17.33 ± 0.6 | 31.5 ± 1.53 |
| 75% AW | 9.23 ± 0.31 | 2.79 ± 0.08 | 2.4 ± 0.07 | 3.43 ± 0.16 | 8.78 ± 0.91 | 7.89 ± 1.02 | 18.78 ± 0.94 | 21 ± 0.58 | 36.17 ± 0.73 |
| 50% AW | 8.59 ± 0.33 | 1.24 ± 0.13 | 2.48 ± 0.06 | 3.66 ± 0.08 | 8.87 ± 0.38 | 7.88 ± 0.06 | 15 ± 0.86 | 19.33 ± 1.09 | 31.83 ± 1.17 |
| 25% AW | 7.44 ± 0.06 | 0.55 ± 0.07 | 2.49 ± 0.07 | 3.39 ± 0.14 | 7.21 ± 0.74 | 6.07 ± 0.97 | 12.88 ± 0.74 | 17.67 ± 0.17 | 28.33 ± 0.33 |
| **UTLA9** | | | | | | | | | |
| 100% AW | 8.96 ± 0.13 | 1.43 ± 0.13 | 2.31 ± 0.05 | 3.38 ± 0.12 | 12.81 ± 0.78 | 11.72 ± 0.64 | 23.25 ± 0.59 | 17.33 ± 0.88 | 36.5 ± 1.32 |
| 75% AW | 8.74 ± 0.41 | 2.27 ± 0.18 | 2.54 ± 0.07 | 3.84 ± 0.11 | 10.74 ± 0.9 | 10.3 ± 0.71 | 21.66 ± 1.28 | 18.5 ± 0.29 | 36.17 ± 1.88 |
| 50% AW | 6.92 ± 0.36 | 1.16 ± 0.14 | 2.42 ± 0.09 | 3.71 ± 0.04 | 10.92 ± 0.84 | 9.19 ± 0.91 | 19.87 ± 0.13 | 16.5 ± 1.04 | 34.67 ± 0.67 |
| 25% AW | 8.96 ± 0.14 | 1.91 ± 0.02 | 2.18 ± 0.05 | 2.98 ± 0.06 | 8.56 ± 0.29 | 6.73 ± 0.91 | 15.67 ± 1.18 | 14 ± 0.5 | 30.5 ± 1.76 |
| **UTLA10** | | | | | | | | | |
| 100% AW | 8 ± 0.19 | 1.19 ± 0.08 | 2.37 ± 0.06 | 3.67 ± 0.04 | 9 ± 0.96 | 8.89 ± 0.87 | 17.2 ± 0.31 | 18.17 ± 0.88 | 31.5 ± 1 |
| 75% AW | 8.83 ± 0.33 | 1.32 ± 0.05 | 2.46 ± 0.08 | 3.67 ± 0.08 | 10.97 ± 0.51 | 9.72 ± 0.36 | 17.63 ± 0.59 | 18.83 ± 0.93 | 33.17 ± 0.93 |
| 50% AW | 8.44 ± 0.48 | 1.31 ± 0.13 | 2.39 ± 0.04 | 3.49 ± 0.09 | 8.78 ± 1.61 | 7.69 ± 0.54 | 14 ± 0.51 | 20.43 ± 0.74 | 30.17 ± 1.48 |
| 25% AW | 9.05 ± 0.22 | 1.48 ± 0.06 | 2.42 ± 0.01 | 3.86 ± 0.16 | 7.78 ± 0.22 | 6.41 ± 0.13 | 14.72 ± 1.07 | 17.83 ± 0.44 | 29.67 ± 1.3 |
| **UTLA12** | | | | | | | | | |
| 100% AW | 8.39 ± 0.29 | 1.24 ± 0.1 | 2.39 ± 0.04 | 3.81 ± 0.15 | 9.58 ± 1.05 | 9.4 ± 0.97 | 18.97 ± 0.8 | 19.67 ± 0.88 | 36.17 ± 0.73 |
| 75% AW | 8.72 ± 0.24 | 1.38 ± 0.29 | 2.38 ± 0.07 | 3.93 ± 0.13 | 10.22 ± 0.67 | 9.44 ± 0.73 | 14.17 ± 0.73 | 13.33 ± 0.6 | 22.17 ± 1.01 |
| 50% AW | 8.86 ± 0.56 | 0.45 ± 0.06 | 2.46 ± 0.08 | 3.91 ± 0.08 | 7.42 ± 0.3 | 6.83 ± 0.17 | 16.92 ± 0.92 | 15.33 ± 0.88 | 28 ± 1.53 |
| 25% AW | 8.34 ± 0.6 | 0.59 ± 0.07 | 2.41 ± 0.07 | 3.68 ± 0.07 | 7.02 ± 0.39 | 6.11 ± 0.36 | 13.88 ± 0.51 | 15.83 ± 0.6 | 25.83 ± 1.42 |
| LSD 1% | 1.5 | 0.44 | 0.2 | 0.47 | 3.05 | 2.95 | 2.68 | 2.96 | 4.67 |
| **UTLA1** | | | | | | | | | |
| 100% AW | 16.07 ± 1.27 | 5.33 ± 0.33 | 1.44 ± 0.29 | 34 ± 0.58 | 26.6 ± 0.81 | 21.85 ± 1.47 | 5.92 ± 0.42 | 24.11 ± 3.2 | 3.44 ± 0.32 |
| 75% AW | 19.08 ± 0.58 | 3 ± 0.58 | 1.56 ± 0.06 | 32.33 ± 0.67 | 23.5 ± 0.87 | 27.4 ± 0.35 | 7.68 ± 0.19 | 46.05 ± 2.42 | 5.19 ± 0.67 |
| 50% AW | 20.07 ± 0.23 | 2.33 ± 0.33 | 2.67 ± 0.17 | 34.33 ± 0.6 | 27.5 ± 0.87 | 17.38 ± 0.36 | 4.46 ± 0.86 | 21.17 ± 4.2 | 2.32 ± 0.62 |
| 25% AW | 19.98 ± 0.59 | 2 ± 0.58 | 4.22 ± 0.4 | 30.17 ± 0.44 | 32 ± 0.58 | 10 ± 1.15 | 3.7 ± 0.17 | 20.61 ± 0.8 | 3.05 ± 0.03 |

**Table 3.** *Cont.*

| Accession/ Irrigation | Number of Seeds per Capsule | Seed Yield per Plant (g) | Seed Width (mm) | Weight of 1000 Seeds (g) | Number of Flowers in Inflorescence | Number of Mature Capsules | Inflorescence Length (cm) | Length of the Main Branch (cm) | Plant Height (cm) |
|---|---|---|---|---|---|---|---|---|---|
| **UTLA6** | | | | | | | | | |
| 100% AW | 18.45 ± 1.12 | 5.83 ± 0.44 | 1.39 ± 0.46 | 32.78 ± 0.62 | 25 ± 0.58 | 28.19 ± 0.73 | 10.41 ± 0.43 | 56.9 ± 5.12 | 6.33 ± 0.88 |
| 75% AW | 21.97 ± 0.38 | 4.67 ± 0.88 | 1.22 ± 0.22 | 34.56 ± 0.68 | 27.05 ± 1.18 | 39.95 ± 1.7 | 14.05 ± 1.18 | 48.35 ± 5.98 | 8.1 ± 0.64 |
| 50% AW | 24.33 ± 0.45 | 4 ± 0.58 | 3 ± 0.58 | 31 ± 0.76 | 23.55 ± 0.89 | 26.6 ± 0.58 | 9.62 ± 0.07 | 48.06 ± 6.35 | 8.37 ± 0.21 |
| 25% AW | 21.07 ± 0.52 | 3.67 ± 0.88 | 5.28 ± 0.31 | 33.45 ± 0.48 | 26.57 ± 1.26 | 18.7 ± 0.81 | 7.78 ± 0.05 | 38 ± 4.09 | 5.93 ± 0.16 |
| **UTLA7** | | | | | | | | | |
| 100% AW | 21.19 ± 0.74 | 6.33 ± 0.67 | 1.11 ± 0.4 | 36.87 ± 1.04 | 30.05 ± 1.18 | 56.8 ± 1.27 | 19.44 ± 1 | 62.47 ± 10.91 | 10.56 ± 1.17 |
| 75% AW | 23.78 ± 1.01 | 6.5 ± 0.5 | 1.78 ± 0.22 | 34.11 ± 0.67 | 29 ± 2 | 32.33 ± 1.59 | 13.63 ± 0.89 | 87.05 ± 26.03 | 9.36 ± 2.6 |
| 50% AW | 22.08 ± 1.29 | 4.67 ± 0.67 | 3.56 ± 0.29 | 33.17 ± 1.09 | 24.88 ± 1.07 | 32.8 ± 2.19 | 10.45 ± 0.2 | 56.66 ± 0.81 | 7.91 ± 1.12 |
| 25% AW | 21.36 ± 0.5 | 3.83 ± 0.73 | 4.67 ± 0.38 | 32.72 ± 1.07 | 35.15 ± 0.61 | 23.2 ± 1.33 | 7.85 ± 0.78 | 28.8 ± 0.8 | 5.4 ± 0.29 |
| **UTLA9** | | | | | | | | | |
| 100% AW | 22.32 ± 0.22 | 4.5 ± 0.29 | 1.11 ± 0.11 | 41 ± 0.76 | 33.5 ± 1.44 | 42.03 ± 1.6 | 15.47 ± 1.63 | 65.69 ± 5.31 | 9.03 ± 0.56 |
| 75% AW | 20.9 ± 0.57 | 7.67 ± 0.33 | 1.89 ± 0.11 | 36.08 ± 1.23 | 28.05 ± 0.66 | 44.15 ± 0.66 | 15.91 ± 0.17 | 68.84 ± 5.45 | 11.26 ± 0.86 |
| 50% AW | 20.67 ± 1.15 | 4.67 ± 0.33 | 2.66 ± 0.33 | 38.07 ± 0.95 | 26.55 ± 1.82 | 19.15 ± 2.17 | 6.51 ± 1.03 | 27.23 ± 0.77 | 3.46 ± 0.31 |
| 25% AW | 20.57 ± 1.16 | 4.67 ± 0.33 | 3.78 ± 0.4 | 32.61 ± 0.81 | 30.88 ± 1.77 | 19.6 ± 2.66 | 6.45 ± 0.72 | 27.57 ± 7.34 | 3.79 ± 0.79 |
| **UTLA10** | | | | | | | | | |
| 100% AW | 22.37 ± 0.9 | 3.67 ± 0.67 | 1.22 ± 0.4 | 32.72 ± 0.64 | 32.5 ± 0.87 | 43.76 ± 1.25 | 15.02 ± 1.58 | 91.24 ± 19.24 | 11.02 ± 1.63 |
| 75% AW | 23.48 ± 0.67 | 6.33 ± 0.67 | 1.39 ± 0.06 | 30.43 ± 0.74 | 31 ± 1.15 | 37.28 ± 0.85 | 10.86 ± 0.42 | 56.65 ± 0.4 | 8.02 ± 0.09 |
| 50% AW | 20 ± 0.75 | 5 ± 0.58 | 2.67 ± 0.44 | 35.28 ± 0.55 | 27.25 ± 0.43 | 14.55 ± 0.32 | 4.24 ± 0.14 | 28.45 ± 1.47 | 3.78 ± 0.15 |
| 25% AW | 24.77 ± 1.13 | 5.67 ± 0.67 | 4.17 ± 0.25 | 34.83 ± 0.88 | 25.75 ± 1.24 | 28.62 ± 2.08 | 9.9 ± 0.44 | 58.89 ± 6.21 | 8.92 ± 0.72 |
| **UTLA12** | | | | | | | | | |
| 100% AW | 20.98 ± 0.28 | 4.67 ± 0.67 | 3.11 ± 0.49 | 34.5 ± 1.04 | 27.1 ± 0.52 | 25.78 ± 0.85 | 9.9 ± 0.76 | 38.02 ± 2.21 | 4.15 ± 0.24 |
| 75% AW | 21.68 ± 0.41 | 4.33 ± 0.67 | 3.22 ± 0.28 | 33.39 ± 0.56 | 26 ± 0.58 | 13.2 ± 0.64 | 4.66 ± 0.32 | 35.93 ± 3.89 | 4.53 ± 0.48 |
| 50% AW | 21.83 ± 0.93 | 1.67 ± 0.33 | 2.78 ± 0.29 | 31.5 ± 0.76 | 31.2 ± 0.69 | 13.65 ± 0.95 | 5.13 ± 0.38 | 37.85 ± 3.38 | 4.25 ± 0.37 |
| 25% AW | 22.97 ± 0.88 | 2.67 ± 0.33 | 4 ± 0.51 | 33.55 ± 0.78 | 28.55 ± 0.32 | 30.72 ± 1.36 | 11.31 ± 0.25 | 51.37 ± 1.11 | 6.62 ± 0.73 |
| LSD (1%) | 3.12 | 2.12 | 1.29 | 3.07 | 4.08 | 4.51 | 2.81 | 30.58 | 3.18 |

Values are given as mean ± SE.

In response to reduced irrigation levels, the number of seeds per capsule was either decreased (accessions UTLA1 and UTLA7), increased (accession UTLA10), or not significantly modified (accessions UTLA6, UTLA9, and UTLA12). Seed yield per plant in accessions UTLA1, UTLA7, and UTLA12 were significantly decreased in response to water deficit stress. On the contrary, seed width was not affected by water stress (Table 3). The maximum and minimum seed widths were observed in accessions UTLA7 and UTLA1, respectively. The weight of 1000 seeds and the number of flowers per inflorescence showed a significant decrease only in accession UTLA9. Water deficit led to a significant decrease in the mature capsules number in accessions UTLA9 and UTLA12. The length of the inflorescence decreased in all accessions except for UTLA1, which, interestingly, showed a substantial increase in the length of the inflorescence with an increase in stress. The length of the main branch in UTLA7 and UTLA9 and the height of the plant in UTLA6, UTLA7, and UTLA9 showed a significant decrease under reduced irrigation levels. Although other accessions were not particularly affected by water deficit stress, leaf length in UTLA1 showed a significant increase. By increasing the water deficit stress level, the number of inflorescences per plant in accessions UTLA1, UTLA6, UTLA7, and UTLA12 showed a significant decrease. As expected, the number of chlorotic leaves in all accessions increased significantly with increasing water stress levels. The highest flower diameter was observed in accession UTLA9, although a significant decrease in flower diameter occurred with decreasing irrigation. As the stress level increased, the root length decreased in accession UTLA10 (-26%) and increased significantly in UTLA1 (+20%). By reducing the irrigation level, shoot fresh weight in all accessions except UTLA12 decreased. The shoot dry weight showed a significant decrease in UTLA6, UTLA7, UTLA9, and UTLA10. Root fresh and dry weight in UTLA7 and UTLA9 showed a significant decrease by increasing stress levels (Table 3).

The results indicated that the capsule diameter and length of the seeds and the leaves under water deficit stress did not show significant changes (Table S2). However, a significant difference was observed between accessions. The largest capsule diameter was observed in UTLA12 (5.86 mm) and the lowest in UTLA1 (5.41) (Figure 1). Minimum and maximum seed and leaf lengths were observed in accessions UTLA1 (6.55) and UTLA10 (8.06), respectively (Figure 1).

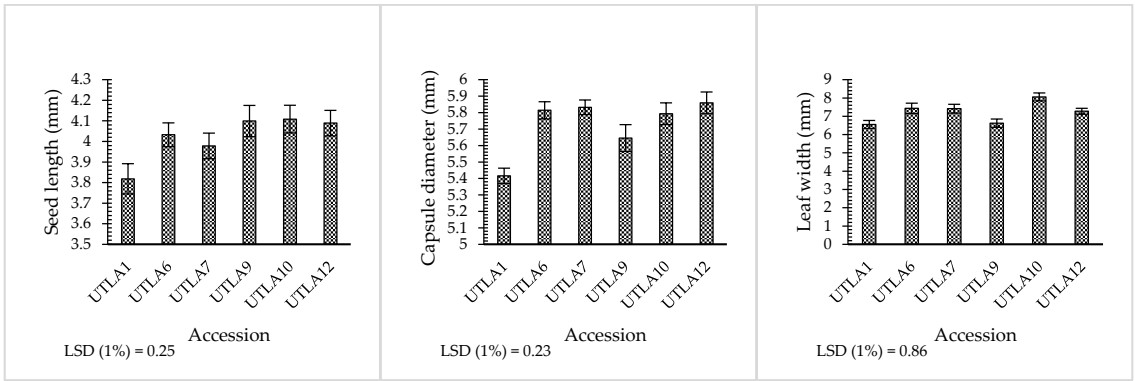

**Figure 1.** Mean of morphological traits in *L. album* accessions. Values are given as mean ± SE.

The capsule-to-flower ratio was affected by water deficit stress, whereas the response of the accessions did not show a significant difference (Table S2). Figure 2 shows the changes in the capsule-to-flower ratio under four irrigation conditions. The lowest percentage of the flower-to-capsule ratio (0.82) with a significant difference compared with other levels was observed at the 25% AW condition.

## 3.2. Phenological Stages

The occurrence of different phenological stages in *L. album* accessions under water deficit stress showed a significant difference (Table S3). The growth of UTLA10 plants began earlier than other accessions (after receiving 47.52 degree days). Plants of accession UTLA12 required more heat units

(64.28) to start growing. Plants of accession UTLA6 entered the bud and flowering stages earlier than other accessions (after receiving 179.90 and 278.45), whereas accession UTLA12 required more heat units to enter these stages (209.54 and 302.71, respectively). Maturation took 648.22 degree days in UTLA10 plants (the longest maturing period), whereas UTLA9 plants (the shortest maturing period) completed their life cycle by receiving 618.82 degree days (Figures 3 and 4).

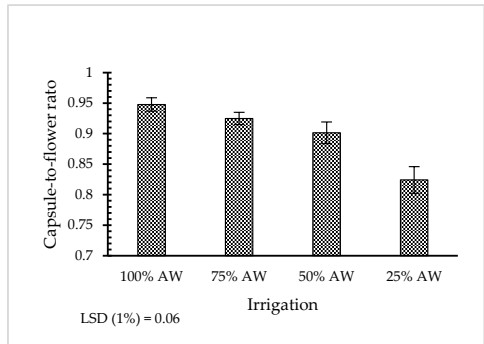

**Figure 2.** Effect of water deficit stresses on the capsule-to-flower ratio of *L. album*. Values are given as mean ± SE.

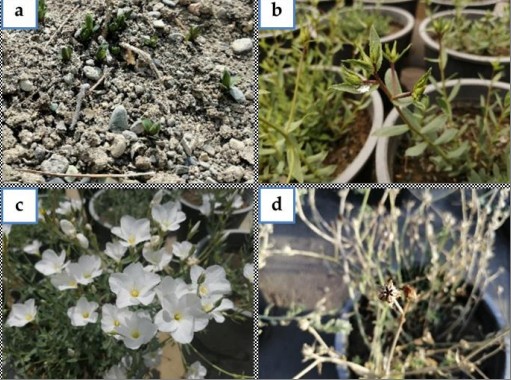

**Figure 3.** Phenological stages in *L. album.* Starting leaf stage (**a**), budding (**b**), flowering (**c**), and maturity (**d**).

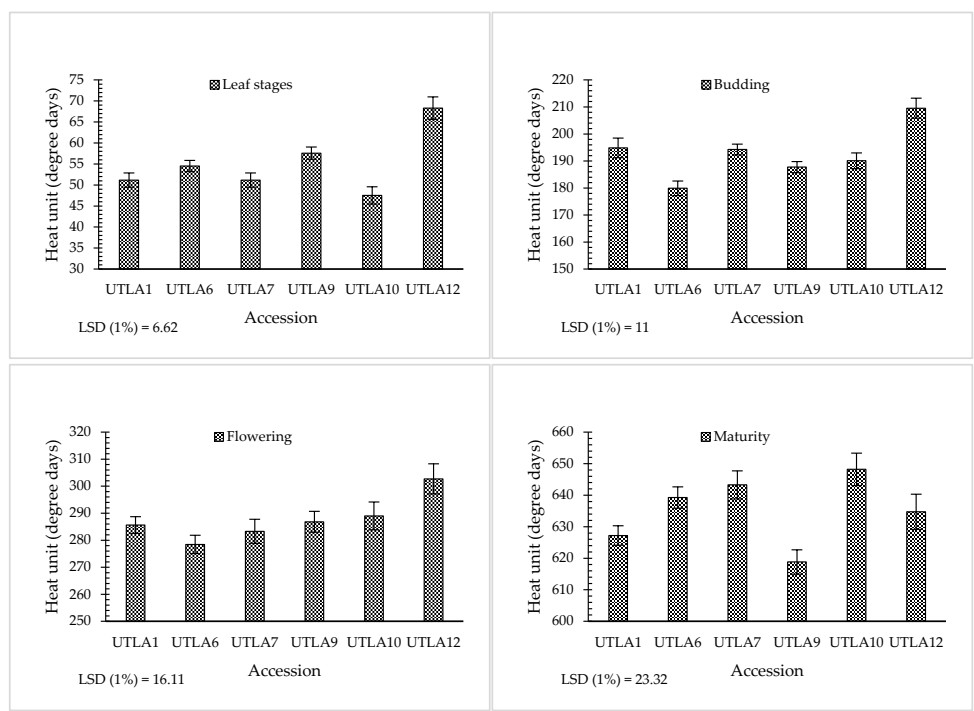

**Figure 4.** Phenological stages of *L. album* accessions. Values are given as mean ± SE.

Among the reported phenological stages, only maturity was affected by water deficit stress (Table S3). Plants under the 25% AW condition matured significantly faster than other levels of irrigation (Figure 5).

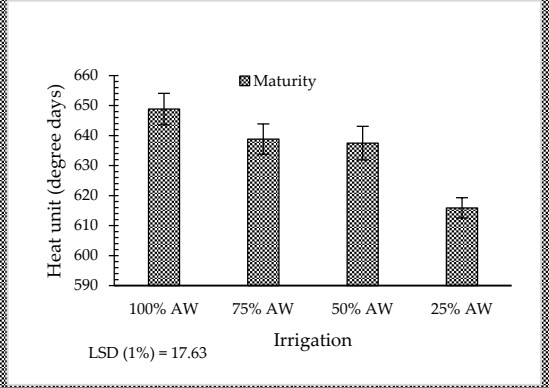

**Figure 5.** Effect of water deficit stresses on the maturity of *L. album*. Values are given as mean ± SE.

### 3.3. RWC, Chlorophyll, and Carotenoid Contents

Based on the results of ANOVA the different accessions showed different physiological responses to irrigation levels (Table S4). In response to the increasing water stress level, the RWC decreased in accessions UTLA6, UTLA7, and UTLA10, increased in UTLA1 and UTLA12, and was observed without any significant change in UTLA9 compared to the control (Table 4). Chlorophyll-a and chlorophyll-b as well as carotenoids showed significant decreases in response to increasing stress levels in all accessions except accessions UTLA6 and UTLA12. Under normal irrigation conditions (control), the highest amount of chlorophyll-a and chlorophyll-b was observed in accession UTLA7, whereas under severe stress, accession UTLA6 showed the highest amounts. Under severe stress (25% AW), accession UTLA1 showed the lowest amount of pigments, whereas accession UTLA7 showed the highest amounts (Table 4).

### 3.4. Proline and Glycine Betaine

A significant increase in proline levels was observed in all accessions in the 25% AW treatment compared to the control. Under the 25% AW treatment, accessions UTLA6 and UTLA7 showed the lowest (4.19 μmol/g FW) and highest (6.01) amount of proline, respectively. With increasing stress levels, the amount of glycine betaine in accessions UTLA1 and UTLA9 showed a significant decrease, whereas an increase was observed for accessions UTLA10 and UTLA12. The highest amount of glycine betaine (249.64 μmol/g DW) was observed in accession UTLA10 under the 25% AW treatment (Table 4 and Table S3).

### 3.5. Electrolyte Leakage and Malondialdehyde

The results revealed that the water deficit stress in accessions UTLA1, UTLA6, UTLA9, and UTLA12 resulted in a significant increase in ion leakage, whereas the ion leakage decreased in accession UTLA10 and there was no significant change in accession UTLA7. In accessions UTLA1, UTLA9, and UTLA10, exacerbation of dehydration caused a significant increase in MDA. Accessions UTLA6 and UTLA12 did not show significant changes (Table 4).

### 3.6. Abscisic Acid

Total ABA content in leaves is reported in Table 4. Increased water stress levels increased the amount of ABA in all accessions with the exception of accession UTLA7. In accession UTLA1, the amount of ABA in the 25% AW treatment was 2.25-fold higher than the control, whereas accessions UTLA6 and UTLA7 did not show significant differences. In accessions UTLA9, UTLA10, and UTLA12, ABA content increased significantly in response to increasing water stress levels.

**Table 4.** Effect of water deficit stresses on physiological traits of *L. album* accessions.

| Accession/ Irrigation | RWC (%) | Chlorophyll-a (µg/mL) | Chlorophyll-b (µg/mL) | Carotenoids (µg/mL) | Proline (µmol/g FW) | Glycine Betaine (µmol/g DW) | Electrolyte Leakage (%) | MDA (µmol/g FW) | ABA (pmol/g DW) |
|---|---|---|---|---|---|---|---|---|---|
| **UTLA1** | | | | | | | | | |
| 100% AW | 77.05 ± 1.98 | 19.3 ± 0.6 | 7.22 ± 0.25 | 5.45 ± 0.15 | 2.64 ± 0.04 | 228.75 ± 7.27 | 55.55 ± 3.71 | 0.102 ± 0.023 | 150.89 ± 3.45 |
| 75% AW | 80.74 ± 0.91 | 16.47 ± 1.48 | 5.51 ± 0.53 | 4.99 ± 0.35 | 5.62 ± 0.3 | 166.48 ± 5.82 | 65.7 ± 1.08 | 0.113 ± 0.0143 | 177.75 ± 11.72 |
| 50% AW | 80.67 ± 2.14 | 17.5 ± 0.19 | 5.55 ± 0.46 | 5.19 ± 0.12 | 3.67 ± 0.23 | 189.09 ± 10.66 | 78.21 ± 2.56 | 0.136 ± 0.0053 | 134.8 ± 13.83 |
| 25% AW | 85.11 ± 1.11 | 13.2 ± 1.82 | 4.72 ± 0.51 | 3.87 ± 0.41 | 5.78 ± 0.15 | 130.19 ± 5.25 | 65.59 ± 2.56 | 0.149 ± 0.005 | 340.01 ± 26.85 |
| **UTLA6** | | | | | | | | | |
| 100% AW | 84.61 ± 0.54 | 20.38 ± 0.27 | 7.43 ± 0.1 | 5.47 ± 0.09 | 4.99 ± 0.22 | 169.59 ± 4.04 | 62.54 ± 0.83 | 0.175 ± 0.0021 | 101.59 ± 5.86 |
| 75% AW | 84.07 ± 0.73 | 19.3 ± 0.19 | 7.01 ± 0.11 | 5.37 ± 0.05 | 5.07 ± 0.21 | 122.69 ± 6.7 | 75.82 ± 0.64 | 0.151 ± 0.0088 | 89.62 ± 5.55 |
| 50% AW | 80.43 ± 0.62 | 19.18 ± 0.33 | 6.93 ± 0.15 | 5.37 ± 0.08 | 4.06 ± 0.32 | 141.62 ± 1.76 | 67.9 ± 1.06 | 0.155 ± 0.0109 | 135.71 ± 13.3 |
| 25% AW | 74.37 ± 1.16 | 20.44 ± 0.63 | 7.59 ± 0.11 | 5.72 ± 0.19 | 4.19 ± 0.05 | 168.04 ± 0.14 | 72.66 ± 1.16 | 0.154 ± 0.0108 | 112.73 ± 1.54 |
| **UTLA7** | | | | | | | | | |
| 100% AW | 84.95 ± 0.92 | 22.68 ± 0.74 | 9.11 ± 0.05 | 5.97 ± 0.19 | 5.28 ± 0.2 | 199.09 ± 6.55 | 64.52 ± 1.89 | 0.184 ± 0.0085 | 136.87 ± 15.44 |
| 75% AW | 88.36 ± 0.94 | 20.88 ± 0.46 | 7.76 ± 0.19 | 6.12 ± 0.14 | 5.22 ± 0.12 | 164.51 ± 5.83 | 74.67 ± 1.28 | 0.12 ± 0.0101 | 137.79 ± 15.53 |
| 50% AW | 86.01 ± 1.1 | 20.15 ± 0.77 | 7 ± 0.32 | 6.47 ± 0.21 | 4.44 ± 0.18 | 152.06 ± 13.22 | 63.27 ± 1.55 | 0.174 ± 0.0102 | 109.44 ± 5.3 |
| 25% AW | 77.38 ± 1.23 | 18.88 ± 0.3 | 6.86 ± 0.36 | 5.76 ± 0.04 | 6.01 ± 0.13 | 209.33 ± 5.82 | 60.9 ± 2.78 | 0.128 ± 0.0099 | 113.19 ± 3.21 |
| **UTLA9** | | | | | | | | | |
| 100% AW | 76.81 ± 0.42 | 21.19 ± 0.1 | 7.75 ± 0.29 | 6.13 ± 0.1 | 5.15 ± 0.08 | 227.11 ± 11.27 | 57.45 ± 1.65 | 0.157 ± 0.0099 | 103.23 ± 6.11 |
| 75% AW | 83.33 ± 1.44 | 21.62 ± 1 | 7.92 ± 0.19 | 5.96 ± 0.09 | 5.04 ± 0.14 | 193.48 ± 2.17 | 75.89 ± 3.42 | 0.139 ± 0.0079 | 90.38 ± 5.29 |
| 50% AW | 82.63 ± 1.25 | 18.99 ± 0.3 | 6.79 ± 0.2 | 5.28 ± 0.07 | 4.38 ± 0.21 | 171.31 ± 8.41 | 71.27 ± 1.53 | 0.208 ± 0.008 | 115.82 ± 4.89 |
| 25% AW | 75.82 ± 1.2 | 17.53 ± 0.33 | 5.81 ± 0.59 | 5.01 ± 0.17 | 5.4 ± 0.15 | 180.33 ± 4.68 | 67.71 ± 1.27 | 0.203 ± 0.014 | 139.39 ± 6.16 |
| **UTLA10** | | | | | | | | | |
| 100% AW | 83.43 ± 0.52 | 21.44 ± 0.56 | 8.03 ± 0.19 | 5.5 ± 0.15 | 4.49 ± 0.14 | 182.79 ± 7.82 | 67.52 ± 1.11 | 0.112 ± 0.0006 | 101.87 ± 1.88 |
| 75% AW | 84.92 ± 0.18 | 20.42 ± 0.61 | 7.43 ± 0.43 | 5.64 ± 0.29 | 5.23 ± 0.17 | 185.98 ± 2.01 | 73.89 ± 1.4 | 0.085 ± 0.0073 | 122.38 ± 4.71 |
| 50% AW | 84.18 ± 1.06 | 17.58 ± 0.47 | 5.73 ± 0.28 | 5.1 ± 0.05 | 5.44 ± 0.16 | 190.73 ± 6.47 | 41.19 ± 3.54 | 0.18 ± 0.0039 | 117.18 ± 6.21 |
| 25% AW | 78.57 ± 0.42 | 16.82 ± 0.13 | 6.25 ± 0.35 | 5.44 ± 0.16 | 5.93 ± 0.15 | 249.64 ± 4.68 | 58.36 ± 0.9 | 0.182 ± 0.0066 | 155.81 ± 11.55 |
| **UTLA12** | | | | | | | | | |
| 100% AW | 83.59 ± 2.92 | 17.58 ± 0.79 | 6.1 ± 0.32 | 5.25 ± 0.23 | 4.5 ± 0.1 | 168.78 ± 4 | 60 ± 1.85 | 0.131 ± 0.0079 | 117.36 ± 9.29 |
| 75% AW | 85.46 ± 0.92 | 17.14 ± 0.67 | 6.17 ± 0.09 | 4.64 ± 0.17 | 3.21 ± 0.05 | 215.52 ± 7.89 | 67.95 ± 1.57 | 0.122 ± 0.009 | 150.82 ± 11.34 |
| 50% AW | 87.5 ± 1.24 | 15.28 ± 0.65 | 5.58 ± 0.21 | 4.69 ± 0.19 | 5.72 ± 0.16 | 215.93 ± 8.4 | 80.17 ± 0.35 | 0.146 ± 0.0067 | 204.98 ± 10.91 |
| 25% AW | 82.18 ± 1.32 | 18.8 ± 0.48 | 6.72 ± 0.1 | 5.87 ± 0.14 | 5.46 ± 0.08 | 195.9 ± 7.14 | 85.14 ± 3.61 | 0.14 ± 0.0068 | 174.2 ± 7.06 |
| LSD (1%) | 4.536 | 2.011 | 1.156 | 0.562 | 0.658 | 25.7 | 7.768 | 0.0375 | 37.79 |

Values are given as mean ± SE.

### 3.7. Enzymatic Antioxidant Activity

Figure 6 presents the enzymatic antioxidant activity of *L. album* accessions. In response to a decreased irrigation level, the catalase enzyme activity increased (+76.9%) in accession UTLA1, whereas it showed a significant decrease in accessions UTLA6 (−107%), UTLA7 (−133%), and UTLA10 (−142%) (Figure 6). The guaiacol peroxidase enzyme activity of *L. album* accessions showed substantial variation in response to the different irrigation levels (Table S4). Accession UTLA1 showed the highest guaiacol peroxidase enzyme activity, whereas UTLA7 showed the lowest activity. The activity of this enzyme also showed a significant increase with increasing stress levels in accessions UTLA6 and UTLA10. The response of different accessions based on the ascorbate peroxidase enzyme activity showed a significant difference. Decreases were observed in accessions UTLA7 and UTLA10, whereas significant increases were observed in accessions UTLA9 and UTLA12 (Figure 6).

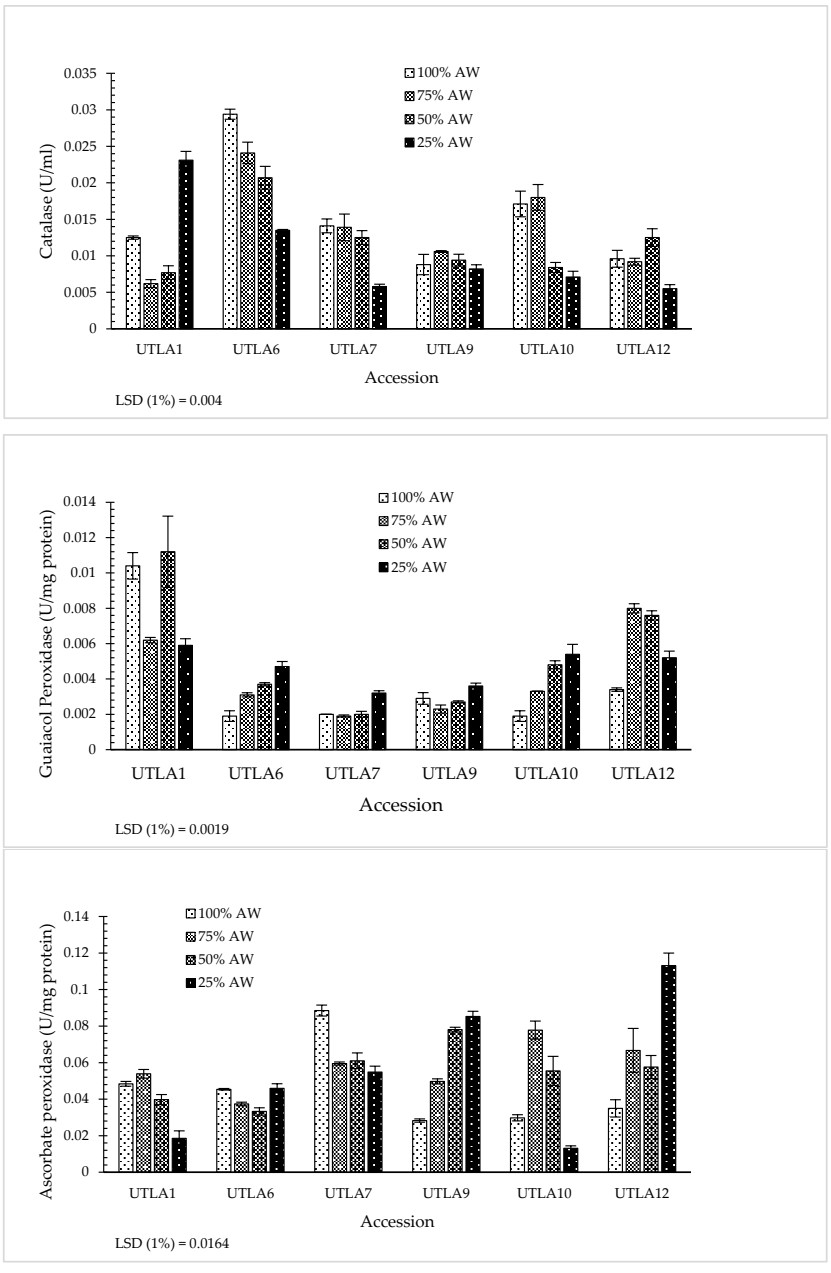

**Figure 6.** Effect of water deficit stresses on the enzymatic antioxidant activity of *L. album* accessions. Values are given as mean ± SE.

### 3.8. Correlations between Traits

Significant correlations were found between the studied traits. A positive correlation coefficient of the number of inflorescences with seed yield ($r = 0.81$), the number of mature capsules ($r = 0.96$), seed length with leaf length ($r = 0.65$), weight of 1000 seeds ($r = 0.72$), correlations of shoot dry weight with root dry weight ($r = 0.90$), chlorophyll-a and chlorophyll-b with the number of inflorescences ($r = 0.65$ and 0.64), shoot dry weight ($r = 0.79$ and 0.83), and root dry weight ($r = 0.72$ and 0.77) were observed. Guaiacol peroxidase activity with plant height ($r = -0.72$) and shoot dry weight ($r = -0.70$), ABA content with chlorophyll-a, chlorophyll-b, and carotenoids ($r = -0.75$, $-0.60$, and $-0.69$) were negatively correlated (Table S5).

### 3.9. Cluster Analysis and Factor Analysis

Accessions were divided into three different clusters based on hierarchical clustering analysis (Figure 7). Accessions UTLA6, UTLA10, and UTLA12 were placed in a separate group. The accessions of this cluster showed relative response similarity in traits such as inflorescence length, plant height, flower and capsule diameter, seed length, guaiacol peroxidase activity, and maturity. Accessions UTLA7 and UTLA9 showing a similar response (in seed yield per plant, plant height, number of inflorescences per plant, root length, shoot dry weight, chlorophyll-a and chlorophyll-b, carotenoids, seeds per capsule, weight of 1000 seeds, and ascorbate peroxidase activity) were grouped in the same cluster (Figure 7). Accession UTLA1 was placed on its own in a separate cluster, this accession showed the lowest value in most of the measured traits, except for ABA content and guaiacol peroxidase activity.

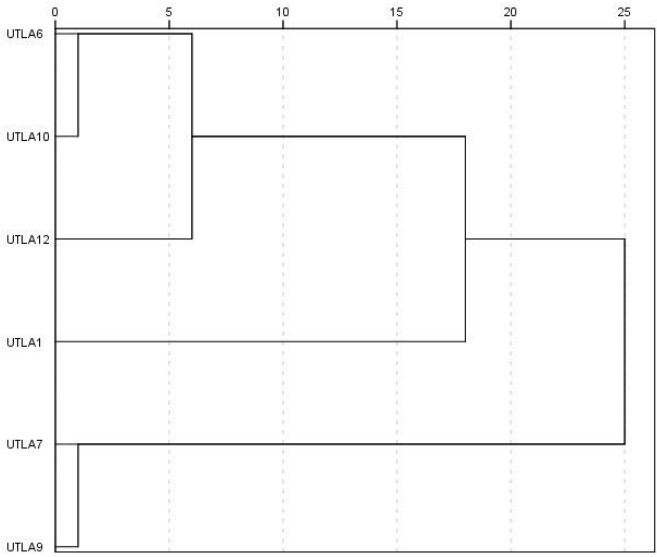

**Figure 7.** Cluster analysis dendrogram of *L. album* accessions.

The factor analysis results of *L. album* accessions showed that five components explain 100% of the total variance (Table S6). The components accounted for 39.07, 18.35, 16.48, 14.19, and 11.92% of the total variance, respectively. The first component was the most important factor in justifying the total variance with traits such as seed yield per plant, length of the main branch, plant height, number of inflorescences per plant, shoot fresh and dry weight, root fresh and dry weight, chlorophyll-a and chlorophyll-b, carotenoids, proline, guaiacol peroxidase, and ABA, with loading factors of 0.84, 0.81, 0.85, 0.91, 0.98, 0.99, 0.87, 0.93, 0.97, 0.98, 0.99, 0.75, $-0.96$, and $-0.83$, respectively (Table S6).

As observed with the hierarchical clustering analysis, accession UTLA1 was placed in a different category in this principal component analysis (Figure 8). The lower performance of this accession in the effective traits of the first component can be considered the main reason for this difference (Table S6). Interestingly, accession UTLA6, with the effective traits in the second component, including catalase

(highest), glycine betaine (lowest), flowering time (earliest), root length (lowest), and the number of seeds per capsule (lowest), was significantly different from other accessions. Accession UTLA7 generally had the highest value in traits related to dry matter production in vegetative organs, such as main branch length, fresh and dry weight of shoots and roots, as well as chlorophyll and proline content. As a consequence, it was isolated from the other accessions in the resulting figure plot (Table S6 and Figure 8).

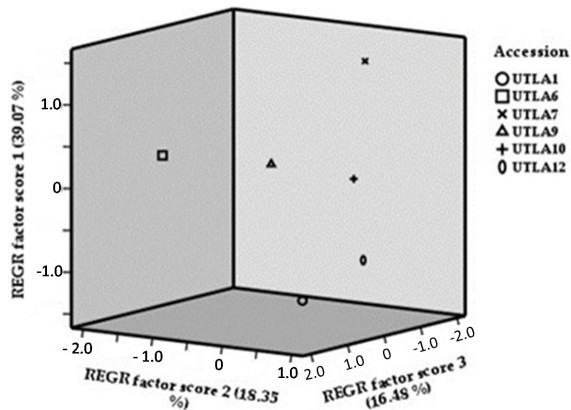

**Figure 8.** Principal component analysis triplot of *L. album* accessions.

## 4. Discussion

In this study, the effect of different levels of water deficit stress on six *L. album* accessions was examined based on various morphological, physiological, and biochemical traits. The results revealed that there was a significant difference between accessions in all the studied traits. However, water deficit, an important factor in plant growth and development, led to different responses in the different accessions. On the basis of the results of the analysis of morphological traits, and given that the experiment was conducted under fully uniform conditions, the high genetic diversity of accessions can be considered the main cause of significant differences, as in previous studies, where the existence of diversity among *L. album* accessions has been stated [13,61,62]. In addition to genetic factors in the occurrence of traits, other explanations for the existence of different responses include the existence of different mechanisms to avoid dehydration stress, including the use of large molecules (mucilage) to adjust the osmotic potential not yet known to wild plant species, such as *L. album*.

Cell growth is among the most drought-sensitive physiological processes due to the reduction in turgor pressure [15]. As revealed by the results of morphological traits, the effect of water deficit stress on accessions was different, and differences were also observed within each accession in their responses under different irrigation levels. For water stress response, severity, duration, and timing of stress, as well as responses of plants after stress removal and the interaction between stress and other factors, are extremely important [63]. Plant growth is typically severely affected by the water deficit. At the morphological level, shoots and roots are the most affected and both are the main components of plant adaptation to drought, as correlated in this study. Plants usually restrict the number and area of leaves in response to drought stress only to reduce water allocation at the expense of loss of yield [64]. As roots are the source of soil water, root growth, density, growth, and size are the main responses of plants to drought stress [65].

Based on the results of this study, the capsule-to-flower ratio in the 25% AW treatment showed a substantial decrease, which may have a serious impact on crop yields. Decreased grain growth in wheat due to reduced sucrose synthase activity [66] and the increased frequency of kernel abortion due to water deficit during pollination in corn [67,68] have been reported. The acceleration of the final stage in seed abortion for plants subjected to this stress tends to be a survival mechanism. Furthermore, the filling period was reduced because of earlier physiological maturity. This shorter filling period reduced seed growth [69]. As a consequence, having different dry matter partitioning patterns in

response to different environmental conditions might vary plant behavior. For example, accession UTLA1 showed the highest seed yield per plant under normal irrigation conditions but the lowest vegetative yield.

A significant effect of the genetic background in *L. album* accessions was observed in the occurrence of different phenological stages, whereas water deficit stress only affected the time of maturity. The effects of drought range from morphological to molecular levels and are evident at any phenological stages. Plants that experienced stress during flower and pod development had a shorter period of organ appearance. This seemed to be due to an increase in the progression rates of reproductive organs under stress. Water stress applied at preanthesis reduced time to anthesis, whereas at postanthesis, it shortened the grain-filling period in triticale genotypes [70]. In summary, plants can escape water deficit stress by shortening their growth duration and avoid stress with the maintenance of high tissue water potential either by reducing water loss from plants or improving water uptake or both. Some plants may reduce their surface area by leaf shedding or producing smaller leaves [24].

Drought stress reduces the relative water content of leaves and is used as a reliable method for measuring the osmotic stress status [71]. RWC is commonly used for the measurement of plant water status in terms of the physiological and biochemical consequences of water deficit in plant cells [72]. It has been reported that plants with higher RWC are more resistant to water deficit stress [73]. In this study, accessions UTLA1 and UTLA12 had the highest RWC under the lowest irrigation level conditions. However, considering the performance traits, we found that these two accessions did not provide acceptable performance. Therefore, RWC cannot be a suitable trait for select tolerant *L. album* accessions (at least for the considered accessions).

Decreased levels of chlorophyll-a and chlorophyll-b have been reported in flaxseed (*Linum usitatissimum*) under dehydration stress [74], in agreement with the decrease in chlorophyll-a, chlorophyll-b, and carotenoid content observed in the *L. album* accessions in the present study. In most plant species, water deficit stress decreases the level of chlorophyll-a, chlorophyll-b, and total chlorophyll [18,75,76]. Drought stress induced changes in photosynthetic pigments and components [77], damaged photosynthetic apparatus [22], and decreased activities of the Calvin cycle enzymes, which are important causes of reduced crop yield [78]. In this study, a positive relationship between chlorophyll-a and chlorophyll-b content and yield-related traits (i.e., number of inflorescences, shoot dry weight, and root dry weight) was observed. For example, accession UTLA7 showed a higher chlorophyll content under stress conditions, and subsequently showed the highest vegetative yield (shoot and root dry weight).

Proline, an important compatible solute, is a general response of higher plants, algae, animals, and bacteria to low water potential [79,80]. In plants, its synthesis in leaves at low water potential is caused by a combination of increased biosynthesis and slow oxidation in mitochondria. Despite some controversy, many physiological roles have been assigned to free proline. Proline is considered a stabilizer of macromolecules, a sink for excess reductant, and a store of carbon and nitrogen for use after relief of water deficit [68]. As observed in our study, many experiments have reported an increase in proline levels in plants under water stress conditions in different plants [67,81–84]. Under severe water stress conditions, the amount of proline in accession UTLA7 showed a significant increase. Given the higher vegetative yields of this accession than others, increasing the amount of proline can be seen as a factor to boost plant growth in this accession.

Glycine betaine (GB) was reported to accumulate in response to stress in many plants, including sugar beet (*Beta vulgaris*), spinach (*Spinacia oleracea*), barley (*Hordeum vulgare*), and wheat (*Triticum aestivum*) [85–87]. In these species, tolerant genotypes accumulate more GB than sensitive genotypes in response to water stress. This, however, is not a general association, and it is most likely that the relationship between GB accumulation and stress tolerance is species- or even genotype-specific. As in *L. album*, the response of each accession was different. Here, with increasing stress levels,

the amount of GB increased in accession UTLA10 but decreased in accessions UTLA1 and UTLA9. Therefore, this trait cannot be considered decisive for *L. album* drought tolerance.

The determination of MDA concentration, a membrane lipid peroxidation product, is used for quantifying the level of membrane peroxidation that leads to ion leakage [88]. Under stress conditions, the unsaturated fatty acids of the cell membranes are impressed by free radicals and form a chain reaction of lipid peroxidation [89]. In this study, ion leakage and MDA content were relatively low in plants under normal irrigation treatment but increased as water stress intensified in accessions UTLA1 and UTLA9, indicating the loss of cell stability and viability [90,91]. Zarrinabadi et al. [83] reported increased MDA content in pot marigold (*Calendula officinalis*) genotypes under water deficit conditions. Stress = sensitive species exhibit a sharper increase in lipid peroxidation than regular species under water deficit stress [92,93]. Here, ion leakage decreased in accession UTLA10 in response to the increasing water stress level, whereas no significant decrease in MDA was observed in other accessions. Since MDA is the final product of lipid peroxidase, various studies have reported that lower MDA content within genotypes indicates the greater antioxidant activities alongside resistances to arid conditions [94,95].

When plants are faced with water deficit conditions, in most cases, ABA levels increase as a result of increased synthesis [96]. A variation in the response of *L. album* accessions to water deficit stress based on their ABA accumulation was observed in the present study. Accessions UTLA2 and UTLA3 did not show any significant change in their ABA content under different irrigation levels, whereas in accessions UTLA1, UTLA4, UTLA,5 and UTLA6, ABA content was significantly increased. This increase was even more pronounced in accession UTLA1, and as a result, its root length was significantly higher after the 25% AW treatment. ABA affects the relative growth rates of different parts of the plant, such as raising the root-to-shoot dry weight ratio, inhibiting the development of the leaf area and growing deeper roots [97]. Here, we reported that the amount of ABA showed a negative correlation with pigments (chlorophyll-a, chlorophyll-b, and carotenoids) content. Under water stress conditions, apoplastic pH increases resulting in increased retention of ABA, serving as a signal for stomatal closure, accompanied by a reduction of transpiration in leaves, which is a significant water conservation response [52,98].

In the present study, the activity of antioxidant enzymes varied as a function of both the genetic background (i.e., accessions) and stress levels (i.e., %AW). A variation in the response mechanism of the antioxidant enzyme activity was clearly observed. ROS production is affected by the severity and duration of stress, species, genotype, and the developmental stage of the plant [99,100]. It is also affected by the ability of the plant to adapt to stress conditions [101]. When the defense response is unable to neutralize high levels of ROS, oxidative stress occurs. In this study, the response of accession UTLA1 to increasing water stress level was an increase in CAT activity, whereas APX activity decreased. However, in accessions UTLA6 and UTLA10, CAT activity decreased and POX activity increased. Interestingly, accession UTLA12 showed a decrease in CAT activity and an increase in APX activity. Catalase is the main antioxidant enzyme that scavenges the oxidant $H_2O_2$ by decomposition to oxygen and water [102]. In drought-tolerant genotypes, catalase activity increased under water stress conditions, which could be an adaptive mechanism to ROS [103]. Ascorbate peroxidase is a key antioxidant enzyme in plants [104], which catalyzes the conversion of $H_2O_2$ into $H_2O$. It has been reported that ascorbate peroxidase activity increases alongside other enzymes under drought stress [105] Askari and Ehsanzadeh [81] studied different drought treatments on fennel genotypes, and reported a higher increase of antioxidant enzymes, in particular catalase, in drought-tolerant genotypes. But in our research, and at least in accession UTLA1, this does not seem to be the case. The same effect of water deficit on the antioxidant content has been illustrated in medicinal plants such as peppermint (*Mentha piperita*), Oregano (*Origanum vulgare*), and marigold genotypes [83,106,107].

Based on the results of factor analysis, it is possible to determine the most effective variables to identify more tolerant accessions in future studies. Due to the placement of most of the influential

traits for the vegetative and reproductive yield in *L. album*, the first PCA component can be considered a yield-influencing factor.

## 5. Conclusions

Evaluating the response of plant accessions under different environmental conditions can help to understand the growth characteristics of plants. Overall, significant diversity was observed in the different *L. album* accessions, which is very important for the advancement of breeding programs. A decrease in the response to increased water deficit stress was observed in most of the morphological traits studied here. Under normal irrigation conditions, accession UTLA1 is the best for seed production, and accessions UTLA7 and UTLA10 are better for the production of vegetative sections. Under severe stress conditions, however, UTLA9 appears to be more suitable for seed production, whereas UTLA10 and UTLA12 may be used for the production of vegetative parts. In general, accessions UTLA7, UTLA9, and UTLA10 had higher vegetative and seed yields and therefore are recommended for use in future breeding research. Water deficit stress accelerates the maturation of the plant, and accession UTLA9 completes its growth cycle faster; therefore, if the duration of the growth period is an important factor for production, this accession can be considered an ideal option for seed production. On the contrary, if the vegetative yield performance during the flowering stage is examined, accession UTLA6 may be recommended due to early entry into the flowering stage. Genetic variation between accessions was also confirmed by physiological responses. The examination of physiological characteristics confirms the enhanced efficiency of accessions UTLA7 and UTLA9. Overall, accession UTLA1 showed unique characteristics and responses among the accessions, whereas accessions UTLA7 and UTLA9 were almost identical in their response. The use of leaf length traits and chlorophyll content to estimate seed yield and shoot yield may be an appropriate marker. Traits such as seed yield per plant, plant height, number of inflorescences per plant, shoot and root dry weight, chlorophyll-a and chlorophyll-b, carotenoids, proline, guaiacol peroxidase, and ABA were obtained as important and discriminant traits for identifying tolerant *L. album* accessions. According to this analysis, it is recommended to use morphological traits to distinguish higher accessions.

To summarize, different accessions were studied and proposed for the advancement of breeding programs. Here, the productivity of *L. album* plants tends to be influenced by genotype rather than water deficit stress. However, we note that the water requirements of the *L. album* accessions also tend to be essential to the optimum achievement of some agronomical traits.

**Supplementary Materials:** The following are available online at http://www.mdpi.com/2073-4395/10/12/1966/s1. Table S1—Daily temperature conditions of field; Table S2—Analysis of variance of morphological traits; Table S3—Analysis of variance of phenological traits; Table S4—Analysis of variance of physiological traits; Table S5—Correlation between traits by Pearson method; Table S6—Factor analysis matrix of *L. album* accessions.

**Author Contributions:** Conceptualization: V.N., M.S.; data curation: R.K.; formal analysis: M.S. and R.K.; funding acquisition: V.N., M.S. and C.H.; investigation: R.K.; methodology: V.N., M.S., C.H. and R.K.; project administration: V.N. and M.S.; resources: V.N., M.S., C.H. and R.K.; software: R.K.; supervision; V.N. and M.S.; validation: V.N., M.S., C.H. and R.K.; visualization: R.K.; writing—original draft: R.K.; writing—review and editing: V.N., M.S., C.H. and R.K. All authors have read and agreed to the published version of the manuscript.

**Funding:** This research was supported by grants from University of Tehran and Région Centre-Val de Loire (Acti-LIN).

**Acknowledgments:** R.K. acknowledges research fellowship from Ministry of Science, Research and Technology of Iran.

**Conflicts of Interest:** The authors declare no conflict of interest.

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
