# Peer review of "Morphological, Physiological, and Biochemical Impacts of Different Levels of Long-Term Water Deficit Stress on Linum album Ky. ex Boiss. Accessions"

_agronomy, doi:10.3390/agronomy10121966_

Round 1

Reviewer 1 Report

Strength: The study is a complete set of phenotyping addressing morphological, physiological and biochemical traits at 4 watering conditions in 6 selected accessions Linum album, replicated 3 times with 3 units per replication.

Weakness: The study is so comprehensive in detail that the importance is lost in the report.

Abstract

The abstract need to be reviewed so that it indicate the key findings. Here some findings are over rated while the key findings are ignored. This may be because; the most important traits were not identified. Try to indicate high yielding based on accessions performance at higher and lower watering, if you do this, you will find high seed yield per plant (for e.g.) under stress condition goes to other accessions than mentioned in the abstract.

Introduction

Line 32-61: nicely reviewed the background and importance of the plant

Line 72: Plants employ … remove this sentence, no need to repeat the same idea as Line 71-72

Line 76: consider noun verb agreement

Line 91: remove period before citation

The introduction has no clear problem statement that links with the objective of the paper mentioned, so, the importance of the study in relation to water deficit remains to be justified well.

Material and methods

The research was detail and the methods were sound. I suggest the following for revision:

Indicate whether data including temperature, relative humidity, during the conduct of the research.

It is not clear that whether the pot experiment was conducted in the greenhouse or in the open filed. Make this clear.

How did you manage the 25% watering while 50% of the FC was reported to impose PWP in Table 2? This part is confusing to understand, consider revising.

Line 115: what is leaf mold and what is its purpose? Make it clear for the reader.

Conduct Principal Component Analysis to identify the key traits from your experiment and you may also conduct cluster analysis to see the relationship among your accessions.

Results

You have good sets of results in each parameter. However, the presentation and organization needs to be revised.

You have given too much emphasis for the genotypic effect in most of the reports. You can make your presentation understandable and valuable if you present first the treatment effect, then genotypic effect, followed by the treatment x genotypic interaction. Together with this, whenever possible, interpretation of results is important as you are dealing with multiple traits at a time. This way you can simplify the presentation and make the reading easier to understand.

You have listed many data in tables and graphs, some I found little importance, for e.g Figure 6 is very informative as it indicates and clearly shows the interaction effect as compared to the others. So, graphs and tables wrong way around except Fig. 6.

The result addressed everything that the data provided in which the key traits and findings were not underscored. This makes the manuscript hard to read and understand and identify the important traits and effects. See the recommendation in the material and methods.

Take the ANOVA and correlation tables to Supplemental data, this way you can make your manuscript easier to read.

In tables try to be consistent in number of decimals and reduce the number of digits. For e.g. Table 3 seed yield can be reported in “g” instead of “mg” and the numbers can be reported in 2 decimals (e.g. 1.86±0.03 instead of 1857.67±33.58), while weight of 1000 seeds can be reduced to 1 digit before decimal (e.g. 3.12±0.10 instead of 3124.33±102.63).

Discussion

After taking the above comments, you may also revise the discussion accordingly.

Conclusion

Line 480-484: I suggest this recommendation should be coined in light of water availability and stress.

Author Response

POINTWISE RESPONSE TO REVIEWERS’ COMMENTS ON AGRONOMY-1021464

Dear Editor and Reviewers,

We are grateful for providing us with an opportunity to revise this manuscript.

Thank you for the reviewer comments for agronomy-1021464; they are very helpful and improve the submission.

We have modified the manuscript according to comments and incorporated all of suggestions in it.

Each reviewer comment is indicated below, together with the responses (clarifications/ revisions) we have included. Changes can be visualized on the manuscript version using the tracked changes option (‘with marks’) provided as supplementary file for Reviewers.

Review Report (black) and answers (red)

With kind regards

The Authors

----------------------------------------------------------------------------------------------------------------------------

Reviewer 1

Weakness: The study is so comprehensive in detail that the importance is lost in the report. In revision, we tried to put more emphasis on important traits.

Abstract

The abstract need to be reviewed so that it indicate the key findings. Here some findings are over rated while the key findings are ignored. This may be because; the most important traits were not identified. Try to indicate high yielding based on accessions performance at higher and lower watering, if you do this, you will find high seed yield per plant (for e.g.) under stress condition goes to other accessions than mentioned in the abstract. Necessary changes were made.

Introduction

Line 72: Plants employ … remove this sentence, no need to repeat the same idea as Line 71-72 done

Line 76: consider noun verb agreement done

Line 91: remove period before citation done

The introduction has no clear problem statement that links with the objective of the paper mentioned, so, the importance of the study in relation to water deficit remains to be justified well. Necessary statements was added to the text.

Material and methods

Indicate whether data including temperature, relative humidity, during the conduct of the research. Maximum and minimum temperatures during the experiment were added to the supplementary data.

It is not clear that whether the pot experiment was conducted in the greenhouse or in the open filed. Make this clear. The necessary description was added to the text.

How did you manage the 25% watering while 50% of the FC was reported to impose PWP in Table 2? This part is confusing to understand, consider revising. The irrigation was calculated based on available water (AW=FC-PWP). The necessary description was added to the text.

Line 115: what is leaf mold and what is its purpose? Make it clear for the reader. The necessary description was added to the text.

Conduct Principal Component Analysis to identify the key traits from your experiment and you may also conduct cluster analysis to see the relationship among your accessions. The analysis was performed and the description was added to the paper.

Results

You have good sets of results in each parameter. However, the presentation and organization needs to be revised. The results and discussion were revised.

You have given too much emphasis for the genotypic effect in most of the reports. You can make your presentation understandable and valuable if you present first the treatment effect, then genotypic effect, followed by the treatment x genotypic interaction. Together with this, whenever possible, interpretation of results is important as you are dealing with multiple traits at a time. This way you can simplify the presentation and make the reading easier to understand. The necessary changes were made based on your comments. In some traits, the effect of water stress was not clear, so the direct explanation of its effect was avoided, and instead, the results were expressed using the effect of accessions.

You have listed many data in tables and graphs, some I found little importance, for e.g Figure 6 is very informative as it indicates and clearly shows the interaction effect as compared to the others. So, graphs and tables wrong way around except Fig. 6. We understand your good comments, but unfortunately due to the significant increase in the pages of the paper, if we use graphs for all traits, we did not use this possibility.

The result addressed everything that the data provided in which the key traits and findings were not underscored. This makes the manuscript hard to read and understand and identify the important traits and effects. See the recommendation in the material and methods. By performing the recommended analyzes, more emphasis was placed on important traits.

Take the ANOVA and correlation tables to Supplemental data, this way you can make your manuscript easier to read. done

In tables try to be consistent in number of decimals and reduce the number of digits. For e.g. Table 3 seed yield can be reported in “g” instead of “mg” and the numbers can be reported in 2 decimals (e.g. 1.86±0.03 instead of 1857.67±33.58), while weight of 1000 seeds can be reduced to 1 digit before decimal (e.g. 3.12±0.10 instead of 3124.33±102.63). Necessary changes were made.

Discussion

After taking the above comments, you may also revise the discussion accordingly.  Necessary changes were made.

Conclusion

Line 480-484: I suggest this recommendation should be coined in light of water availability and stress. Necessary changes were made.

Reviewer 2 Report

Review of AGRONOMY Manuscript-1021464

Morphological, physiological and biochemical impacts of different levels of long-term water deficit stress on Linum album Ky. Ex Boiss. accessions

GENERAL COMMENTS

It is a very remarkable manuscript, from authors who have been working with this medicinal plant species and have previously published their related research results. There has long been a keen interest and many research has focused on the examination of the effects of water stress on plant yield, physiology and phytochemicals in agricultural crops including medicinal plants. The manuscript is acceptable with minor revision.

ABSTRACT

L14: correct sentence to: ‘….an important medicinal plant which contains important…….’

INTRODUCTION

L32-33: correct sentence to: ‘This species is an endemic plant of Iran and grows….’

L34: correct sentence to: ‘…this plant lasts from May…….’

L37: correct sentence to: ‘…leading anticancer drug Etoposide…’

L41: correct sentence to: ‘…in wild forests in Asia…….’

L43: correct sentence to: ‘…Now days this species…….’

L49: correct sentence to: ‘…nor extraction from in vitro plant cultures …….’

L57: correct sentence to: ‘…high amounts of podophyllotoxin…….’

L61: correct sentence to: ‘…little is known about the agronomical performance of L. album…….’

L67: correct sentence to: ‘…outcome of water deficit is limiting…….’

L77: correct sentence to: ‘…assimilates during the life…….’

L78: A more up-to-date additional citation is suggested besides ref. 26,27: Nemeskéri et al., 2019. (see below)

L91: correct sentence to: ‘…by free radical stimulation…….’

L98: correct sentence to: ‘…and understand the morphological,…….’

MATERIALS AND METHODS

L103: correct sentence to: ‘This study was a continuation….’

L104: correct sentence to: ‘…of Linum album in the west of Iran…….’

L112: please specify ‘leaf mold’

L115: please specify ‘leaf mold’

RESULTS

L233: correct sentence to: ‘….number of chlorotic leaves….’

L242: correct sentence to: ‘….However, there was a significant….’

L248-251: Table 3. Please improve the layout for better overview and visibility

L257-261: Table 4. Please improve the layout for better overview and visibility

L260: Table 4. Correct heading …Number of chlorotic leaves…….

L295-296: Table 6. Please improve the layout for better overview and visibility

L297-298: Table 7. Please improve the layout for better overview and visibility

L344-347: Table 8. Please improve the layout for better overview and visibility

DISCUSSION

L356-358: Delete from here and incorporate into Introduction sentences starting with ‘Growth is complete through……which are affected by water deficit.’

L372: correct sentence to: ‘….and increased the ….’

L373: correct sentence to: ‘….have been reported….’

L398: correct sentence to: ‘….stress induced changes in….’

L402-405: Delete from here and incorporate into Introduction sentences starting with ‘Another important effect ……and other cellular components…’

L406-409: Delete from here and incorporate into Introduction sentences starting with ‘The reactive oxygen species ……and amino acids…’

L417-422: Delete from here and incorporate into Introduction sentences starting with ‘Glycine betaine is another ……in signal transduction pathways..’

L418: correct sentence to: ‘….glycine betaine plays ….’

L419: correct sentence to: ‘….under many abiotic stresses….’

L427: correct sentence to: ‘….As in L. album, the ….’

L443: correct sentence to: ‘….Plants respond to water deficit ….’

L443-445: Delete from here and incorporate into Introduction sentences starting with ‘Plants respond to water deficit ……and other environmental stresses…’

L454: correct sentence to: ‘….results section, we have reported that the amount ….’

L456: correct sentence to: ‘….increases resulting in more ….closure and then followed…’

L461-464: Delete from here and incorporate into Introduction sentences starting with ‘Major enzymes of the enzymatic antioxidant ……glutathione reductase (GR)…’

CONCLUSIONS

L493: correct sentence to: ‘……relationship in different accessions can yield……..’

L494: correct sentence to: ‘……easily measurable trait is recommended for the screening of tolerant genotypes.…….

L494-495: correct sentence to: ‘……Finally, it can be said that the yield of L. album plants was more affected by the genotype…….’

REFERENCES

Please check for use of  ,  or  ;  to separate names of authors throughout the Refences section to be uniform according to requirements laid down in instructions for authors.

L509: reference 1.: please add page numbers

L518-520: reference 5.: correct list of authors … add page numbers

L526: reference 8.: correct list of authors…

L538: reference 13.: please add page numbers

L541: reference 15.: please add page numbers

L560: reference 24.: please add page numbers

L568: reference 28.: please add page numbers

L570: reference 29.: please add page numbers

L575: reference 31.: please add page numbers

L600: reference 40.: please add page numbers

L629: reference 52.: please add page numbers

L631: reference 53.: please add page numbers

L632: reference 54.: please add page numbers

L633: reference 55.: please add page numbers

Suggested additional citation:

Nemeskéri, E. et al. Physiological factors and their relationship with the productivity of processing tomato under different water supplies. Water 2019, 11(3) Paper:586, 15.

Author Response

POINTWISE RESPONSE TO REVIEWERS’ COMMENTS ON AGRONOMY-1021464

Dear Editor and Reviewers,

We are grateful for providing us with an opportunity to revise this manuscript.

Thank you for the reviewer comments for agronomy-1021464; they are very helpful and improve the submission.

We have modified the manuscript according to comments and incorporated all of suggestions in it.

Each reviewer comment is indicated below, together with the responses (clarifications/ revisions) we have included. Changes can be visualized on the manuscript version using the tracked changes option (‘with marks’) provided as supplementary file for Reviewers.

Review Report (black) and answers (red)

With kind regards

The Authors

----------------------------------------------------------------------------------------------------------------------------

Reviewer 2

ABSTRACT

L14: correct sentence to: ‘….an important medicinal plant which contains important…….’ done

 INTRODUCTION

L32-33: correct sentence to: ‘This species is an endemic plant of Iran and grows….’ done

L34: correct sentence to: ‘…this plant lasts from May…….’ done

L37: correct sentence to: ‘…leading anticancer drug Etoposide…’ done

L41: correct sentence to: ‘…in wild forests in Asia…….’ done

L43: correct sentence to: ‘…Now days this species…….’ done

L49: correct sentence to: ‘…nor extraction from in vitro plant cultures …….’ done

L57: correct sentence to: ‘…high amounts of podophyllotoxin…….’ done

L61: correct sentence to: ‘…little is known about the agronomical performance of L. album…….’ done

L67: correct sentence to: ‘…outcome of water deficit is limiting…….’ done

L77: correct sentence to: ‘…assimilates during the life…….’ done

L78: A more up-to-date additional citation is suggested besides ref. 26,27: Nemeskéri et al., 2019. (see below) done

L91: correct sentence to: ‘…by free radical stimulation…….’ done

L98: correct sentence to: ‘…and understand the morphological,…….’ done

 MATERIALS AND METHODS

L103: correct sentence to: ‘This study was a continuation….’ done

L104: correct sentence to: ‘…of Linum album in the west of Iran…….’ done

L112: please specify ‘leaf mold’ done

L115: please specify ‘leaf mold’ The necessary description was added to the text.

 RESULTS

L233: correct sentence to: ‘….number of chlorotic leaves….’ done

L242: correct sentence to: ‘….However, there was a significant….’ done

L248-251: Table 3. Please improve the layout for better overview and visibility Changes were made.

L257-261: Table 4. Please improve the layout for better overview and visibility Changes were made.

L260: Table 4. Correct heading …Number of chlorotic leaves……. done

L295-296: Table 6. Please improve the layout for better overview and visibility The table was moved to the supplementary section based on other reviewer comments.

L297-298: Table 7. Please improve the layout for better overview and visibility Changes were made.

L344-347: Table 8. Please improve the layout for better overview and visibility The table was moved to the supplementary section based on other reviewer comments.

  DISCUSSION

L356-358: Delete from here and incorporate into Introduction sentences starting with ‘Growth is complete through……which are affected by water deficit.’ done

L372: correct sentence to: ‘….and increased the ….’ done

L373: correct sentence to: ‘….have been reported….’ done

L398: correct sentence to: ‘….stress induced changes in….’ done

L402-405: Delete from here and incorporate into Introduction sentences starting with ‘Another important effect ……and other cellular components…’ done

L406-409: Delete from here and incorporate into Introduction sentences starting with ‘The reactive oxygen species ……and amino acids…’ done

L417-422: Delete from here and incorporate into Introduction sentences starting with ‘Glycine betaine is another ……in signal transduction pathways..’ done

L418: correct sentence to: ‘….glycine betaine plays ….’ done

L419: correct sentence to: ‘….under many abiotic stresses….’ done

L427: correct sentence to: ‘….As in L. album, the ….’ done

L443: correct sentence to: ‘….Plants respond to water deficit ….’ done

L443-445: Delete from here and incorporate into Introduction sentences starting with ‘Plants respond to water deficit ……and other environmental stresses…’ done

L454: correct sentence to: ‘….results section, we have reported that the amount ….’ done

L456: correct sentence to: ‘….increases resulting in more ….closure and then followed…’ done

L461-464: Delete from here and incorporate into Introduction sentences starting with ‘Major enzymes of the enzymatic antioxidant ……glutathione reductase (GR)…’ done

  CONCLUSIONS

L493: correct sentence to: ‘……relationship in different accessions can yield……..’ done

L494: correct sentence to: ‘……easily measurable trait is recommended for the screening of tolerant genotypes.……. done

L494-495: correct sentence to: ‘……Finally, it can be said that the yield of L. album plants was more affected by the genotype…….’ done

  REFERENCES

Please check for use of  ,  or  ;  to separate names of authors throughout the Refences section to be uniform according to requirements laid down in instructions for authors.

L509: reference 1.: please add page numbers done

L518-520: reference 5.: correct list of authors … add page numbers done

L526: reference 8.: correct list of authors… done

L538: reference 13.: please add page numbers done

L541: reference 15.: please add page numbers done

L560: reference 24.: please add page numbers done

L568: reference 28.: please add page numbers done

L570: reference 29.: please add page numbers done

L575: reference 31.: please add page numbers done

L600: reference 40.: please add page numbers This is a thesis and we followed the instructions.

L629: reference 52.: please add page numbers This is a Conference paper and we followed the instructions (We used the available information)

L631: reference 53.: please add page numbers This is a Conference paper and we followed the instructions (We used the available information)

L632: reference 54.: please add page numbers done

L633: reference 55.: please add page numbers done

Suggested additional citation:

Nemeskéri, E. et al. Physiological factors and their relationship with the productivity of processing tomato under different water supplies. Water 2019, 11(3) Paper:586, 15. Was added to the list of references (reference N.31).

Reviewer 3 Report

The paper is well written and the results reported in there are interesting and worthy of publication. However, I've some concerns and suggestions with the hope to improve the quality of the paper and make it beneficial for a wider audience.

1) Several physiological traits have been evaluated in control plant and in plants grown in 75%, 50% and 25% water deficit. A significant proportion of these traits showed an erratic behiaviour between the different stress intensities. Some times the trait was decreased by 75% water deficit and increased by 25% deficit. Since these puzzling results were found several times, they should be fully discussed. For example, Table 3 : seed yield per plant and genotype UTLA6, UTL7, UTL9,  . Table 3 : plant height , genotype UTLA1.

2) similarly, table 7, electrolytes leakage in genotype UTLA10. more the stress is severe, less there is ion leakage ! how does this could happen ?

3) in the discussion part, the authors listed several traits that were found interesting in discriminating the different genotypes. With this respect and according to the RWC trait, the accessions UTLA1 and UTLA12 should be declared as the most tolerant. However, the authors concluded that the most interesting accessions were UTLA7, 9 and 10. It would be very helpful if the authors propose a sort of classification of the traits according to their pertinence in declaring any accession as truly tolerant.

4) With respect to the suggestion made above,  may be a PCA using a set of non correlated traits and the accessions will give clues to this and provide a synthetic view of how the accessions are close to each other and wich trait is the most discriminant...

Author Response

POINTWISE RESPONSE TO REVIEWERS’ COMMENTS ON AGRONOMY-1021464

Dear Editor and Reviewers,

We are grateful for providing us with an opportunity to revise this manuscript.

Thank you for the reviewer comments for agronomy-1021464; they are very helpful and improve the submission.

We have modified the manuscript according to comments and incorporated all of suggestions in it.

Each reviewer comment is indicated below, together with the responses (clarifications/ revisions) we have included. Changes can be visualized on the manuscript version using the tracked changes option (‘with marks’) provided as supplementary file for Reviewers.

Review Report (black) and answers (red)

With kind regards

The Authors

----------------------------------------------------------------------------------------------------------------------------

Reviewer 3

1) Several physiological traits have been evaluated in control plant and in plants grown in 75%, 50% and 25% water deficit. A significant proportion of these traits showed an erratic behiaviour between the different stress intensities. Some times the trait was decreased by 75% water deficit and increased by 25% deficit. Since these puzzling results were found several times, they should be fully discussed. For example, Table 3 : seed yield per plant and genotype UTLA6, UTL7, UTL9,  . Table 3 : plant height , genotype UTLA1.

2) similarly, table 7, electrolytes leakage in genotype UTLA10. more the stress is severe, less there is ion leakage ! how does this could happen ?

Answer to comments 1 and 2: Your comments are absolutely correct. Due to the fact that L. album is a wild plant, we had predicted that we would see lots of variation among the accessions, so we tried to keep the growing conditions of the plants as uniform as possible (the main reason for culturing the plant in pots). Due to the fact that no data for this plant was reported in many of the measured traits, it was not possible to discuss this in a documented manner. What we think is the high genetic diversity between and within the accessions (mentioned in the paper). Also, the existence of other mechanisms to deal with water stress can justify the irregularity of plant responses (based on our experience, it was observed that there is a significant amount of mucilage in the shoot of this plant, which in itself can be an acceptable mechanism to deal with to justify dehydration, but no case has been reported so far and its need to be tested).

3) in the discussion part, the authors listed several traits that were found interesting in discriminating the different genotypes. With this respect and according to the RWC trait, the accessions UTLA1 and UTLA12 should be declared as the most tolerant. However, the authors concluded that the most interesting accessions were UTLA7, 9 and 10. It would be very helpful if the authors propose a sort of classification of the traits according to their pertinence in declaring any accession as truly tolerant. The necessary explanations were given in the paper. According to the results of other traits, the RWC content was not a suitable trait for selecting the superior accession under water deficit stress.

4) With respect to the suggestion made above,  may be a PCA using a set of non correlated traits and the accessions will give clues to this and provide a synthetic view of how the accessions are close to each other and wich trait is the most discriminant... The analysis was performed and the description was added to the paper.

Round 2

Reviewer 3 Report

Dear Editor
The authors addressed all the questions and suggestions I made.  However, several typo errors are observed and some sentences need to be rewritten for language improvement.  Therefore, I would recommend to accept this version for publication provided that an English editing is done.

sincerely yours

Author Response

Thank you very much. Manuscript has been revised accordingly for typing errors and English. It has been read by 2 colleagues from Leicester University (UK).